# Thermal Effect on Poly(methyl methacrylate) (PMMA) Material Removal in the Micromilling Process

**DOI:** 10.3390/polym12092122

**Published:** 2020-09-17

**Authors:** Ying Yan, Ping Zhou, Huiping Wang, Yu Mao

**Affiliations:** Key Laboratory for Precision and Non-traditional Machining Technology of Ministry of Education, Dalian University of Technology, Dalian 116024, China; pzhou@dlut.edu.cn (P.Z.); whp13804248493@163.com (H.W.); 172375884@mail.dlut.edu.cn (Y.M.)

**Keywords:** thermal effect, PMMA, micromilling, cutting chips

## Abstract

Poly(methyl methacrylate) (PMMA) is of growing interest in the application of microfluidic devices and high precision optical elements due to its excellent moldability and formability. Micromilling is one of the micromachining methods which has been extensively used to manufacture polymer components. In this study, a high-speed micromilling method was used to manufacture polymer with high form accuracy and surface quality. The processing temperature effects on the surface quality were investigated in detail. The dynamic mechanical analysis (DMA) experiment was used to study the material mechanical property under different temperatures. According to the DMA results, the PMMA sample is in the glass and viscoelastic state during the milling process. The cutting chips under various processing temperatures are classified into three kinds according to their shapes: roll, sheet, and sinter. The surface roughness of samples with sheet and roll cutting chips is smaller than that of sinter cutting chips. To obtain a better machining bottom surface and edge shape, the processing temperature below 70 °C is recommended according to the results. This work is of great value for the study of polymer removal mechanism and optimization of processing parameters for the industry.

## 1. Introduction

Polymers such as poly(methyl methacrylate) (PMMA), polyetheretherketone (PEEK), and polytetrafluoroethylene (PTFE) have been widely applied in modern industries due to their distinctive properties, biocompatibility, low thermal conductivity, optical characteristics, and so on [1,2]. Particularly, polymer devices with microstructures are broadly used in many important fields such as precision energy and optical system, microfluidic chips, and drug delivery devices [3,4,5,6]. Therefore, there is a strong demand for miniature polymer devices and components with high form accuracy and surface quality [7,8]. PMMA has good comprehensive mechanical properties, and it is at the forefront of general-purpose plastics. However, it obtains poor heat resistance with the heat distortion temperature of 96 °C [9]. When the temperature is lower than 116 °C, the thermal conductivity of PMMA increases with the increase of temperature, which helps to transfer the heat in machining to the environment in time [10]. 

At present, the main manufacturing methods for PMMA devices are rapid prototyping, injection molding, carbon dioxide (CO_2_) laser etching, photolithography and hot embossing, ultrasonic welding, and ion beam milling [10,11,12]. Although these methods advanced the PMMA manufacturing field, most of these methods require high temperature, complex procedures involving multiple steps, expensive equipment, or chemical processing, which may lead to low productivity, useless polymer melting, and non-uniform polymer compositions [2,13,14,15]. 

The high-speed micromilling is a cheap, time-efficient method able to fabricate the complicated and multi-level microstructures. It is one of the precision machining techniques with a diameter of less than 1 mm. Due to the sharp reduction of tool diameter and feed rate compared to conventional milling, micromilling shows different machining mechanisms than traditional millings, such as minimum cutting thickness, temperature effect, etc., which make the material removal mechanism more complicated [16]. Micromilling of polymer materials is less studied than traditional milling, and a complete theoretical system as well as uniform evaluation standards are in urgent need [17,18].

Some research focused on the machinability characteristics of polymers towards process conditions with the output quality [19,20,21,22,23,24,25,26]. Kobayashi [19] focused on the ultraprecision machining of PMMA. Based on their results, the surface roughness decreased with the feeding rate reduction. They proposed the feasibility of plastic machining with high dimensional accuracy and good surface finish. Albert J. Shih [12] manufactured elastomer with end mills in different diameters found that the machined surface quality was improved by using solid carbon dioxide which indicated that higher temperature was not proper for polymer manufacturing. Emrullah Korkmaz [2] machined the PMMA with a single-crystal diamond end mill cutter with a diameter of 450 μm. In their work, the chip formation, groove integrity, surface roughness, and cutting force were analyzed. They concluded that it was challenging to minimize or eliminate the burr-formation while obtaining high-quality features due to the low glass transition temperature of PMMA. Chris et al. [22] studied micromilling of polystyrene microfluidic channels and concluded that it was difficult to machine polystyrene due to its low elastic modulus and low melting temperature. Xiao et al. [23] found that cutting speed was a critical parameter during macroscale polymer machining due to its viscoelastic property. They also reported that the viscoelastic property of polymers affected the machining process output, and it was challenging work to achieve a good surface finish. Although increasing the spindle speed can achieve better surface roughness, the increased temperatures would lead to the melting of polymers at high spindle speeds. Fetecau et al. [24] also showed that the high cutting speed increased the temperature during macro-scale polymer machining and identified that it was difficult to cut the polymers with low elastic modulus and low melting temperature. Based on the former research about polymer cutting, the process temperature is an essential factor which influences the surface quality a lot. However, the temperature effect is still unclear and further study should be done to reveal the material removal mechanism of polymer materials by machining. In general, the surface quality of the polymer is highly dependent on the processing temperature because of its low glass transition temperature. 

Hence, summarizing some of the reported studies in polymer machining, most of the studies contribute a lot to the relationship between surface quality and processing parameter without considering the characteristics of polymers. As mentioned above, the mechanical properties are sensitive to the change of temperature. It can be concluded that a complete machinability evaluation focused on thermal effects for thermoplastic polymers such as PMMA using a proper design of experiment technique has not been fully addressed. More research should be done to gain a deeper understanding of polymer surface quality influence factor and material removal mechanism to improve the machining quality of polymer in microscale machining. In this work, the research is focused on the thermal effect on PMMA material removal in the micromilling process. The dynamic mechanical analysis (DMA) experiment and the high-speed micromilling full factor experiments were conducted with PMMA samples. The mechanical property, the cutting temperature, the cutting chips property as well as the machined surface quality were studied to analyze the influence of temperature on surface quality. This work may contribute to the process parameter optimization of polymer machining in microscale and ultra-precision machining. 

## 2. Experimental Design and Setup

The main goal of the experimental work is to investigate the effect of temperature during the polymer micromilling process. As is known, the temperature in micromilling process parameters is highly dependent on spindle speed, feed rate, depth of cut, and so on. The material property which can affect the machinability and quality of polymer is highly dependent on the temperature. Therefore, high-speed micromilling is used as the main experimental method. 

The poly (methyl methacrylate) (PMMA) sample employed was a casting bulk sheet in the amorphous state (DX001, Mitsubishi Chemical Polymer, Nantong, China; density: 1.14 g/cm^3^, modulus of elasticity: 3.2 GPa, Vickers hardness of 0.25 GPa) and was 50 mm × 50 mm × 10 mm. The machining process was conducted without using any cutting fluid. A high-precision small micromilling setup was used in this experiment (Figure 1), and it included a high-speed electric spindle (NAKANISHI, E3000, Nakanishi Co., Shanghai, China), three linear precision translation stage (Puai Nano Displacement Technology (Shanghai) Co., Ltd., Shanghai, China, M304) and the infrared camera (IGA 6 Advanced, Lumasense Technologies Co., Stockholm, Sweden). The infrared camera was applied to measure the temperature of the milling zone. According to Sutter et al. [25], the infrared camera can show a heat point near the cutting zone and the temperature near the cutting zone can be calculated based on the material emissivity. Moreover, evaluation and analysis of the different milling cutting parameters, and the influence of temperature generated during milling on material removal, were performed. The ultra-fine particle cemented carbide micromilling cutter (Mitsubishi Japan, purchased from Misumi Co., Shanghai, China, model MS2SS) was applied as the machine tool with a diameter of 0.5 mm, a cutting edge length of 0.75 mm, and a helix angle of 30°. The full factor high-speed micromilling experiments are designed, and the machining process was conducted without using any cutting fluid. The processing parameters were list in Table 1. There are 27 groups of experiments, and each group is repeated three times.

## 3. Results and Discussion

### 3.1. Thermal Effect on PMMA Dynamic Property

In this work, the DMA experiment was carried out with a TA Instruments DMA Q800 Dynamic Mechanical Analyzer (TA instrument Co., New Castle, USA) at a frequency of 1 Hz in the cantilever mode. The experiments were performed on dry samples of prismatic shape. The temperature dependence of storage modulus and loss tangent were studied from 30 °C up to 180 °C with a heating rate of 2 °C /min with a frequency of 1 Hz. The temperature dependence of storage modulus (*E’*) and loss tangent (tan(δ)) were studied from room (25 °C) temperature up to 180 °C with a heating rate of 2 °C /min. 

Dynamic mechanical analysis (DMA) was used to determine the mechanical properties of a viscoelastic material, which is also referred to in this paper as the viscoelastic parameters. These parameters include the storage modulus (*E′*), loss modulus (*E″*), and loss tangent (tan(δ)). The combination of the storage and loss modulus is the dynamic Young modulus (*E**) of the polymer (Equation (1)):(1)$$E^{*}=E^{\prime }+iE^{\prime \prime }$$

The ratio between the storage and loss modulus is the loss factor (tan(δ)) defined in Equation (2). This is a ratio between the dissipated energy and the storage energy per cycle of applied load:(2)$$\tan(\delta )=\frac{E^{\prime \prime }}{E^{\prime }}$$

The analysis of the loss factor tan(δ) presents some advantages over the analysis of the loss modulus. The Young modulus is divided into the loss and storage modulus due to the duality of a viscoelastic response, which is the combination of the elastic and viscous responses of the polymer. The material viscoelastic parameters are temperature dependent. The polymer glass transition is detected in the temperature sweep through a local peak of the loss factor.

The dynamic properties that vary with the temperature of PMMA were shown in Figure 2. The black line is the relationship between tan(δ) and temperature. The red line represents how the storage modulus varies with the temperature. With the increasing temperature, the storage modulus decreases until the temperature is 116 °C and then the storage modulus stays constant. For the tan(δ) curve, the delta(δ) increases until the temperature reaches 116 °C and then decreases. The glass transition temperature *T*g can be found, and it is the peak of the tan(δ) curve, which is 116 °C in this graph. There are three regimes of the tan(δ) curve which is defined as the glassy regime, viscoelastic regime, and rubbery regime. The DMA results also showed that the tan(δ) curve for *T*g increases monotonically in the viscoelastic state (before *T*g) and decreases sharply in the rubber state. The transition from glassy state to viscoelastic state began at 70 °C, which will influence the surface quality during the machining process.

### 3.2. Thermal Effect on Groove Bottom Surface Roughness

The surface roughness of the groove bottom was investigated in this part. A Laser scan confocal microscope (LSCM, VK-X250, Keyence Co., Osaka, Japan) was used to measure the roughness of five points for each groove. The bottom morphology with the best and worst surface roughness Ra was shown in Figure 3, and the surface roughness was in the range of 47 nm and 173 nm in this study. The spindle speed was 20,000 rpm for the cases with different feed rates and cutting depth.

The relationship between feed rate and bottom surface roughness with various spindle speeds was displayed in Figure 4. With the increasing of spindle speed or feed per tooth, the surface roughness changed randomly without any regularity. For the points in Group 1, the surface roughness of these points was about 75 nm, which was very similar to each other among the points. However, the spindle speed of these points was changed from 20,000 rpm to 60,000 rpm. Moreover, the surface roughness of points in Group 2 was about 100 nm with almost the same value of feed per tooth and various spindle speed. Based on the results above, the surface roughness was the comprehensive result of processing parameters and does not change monofonically with any single processing parameter. With the increasing of spindle speed or feed per tooth, more heat would be generated because of the friction between the workpiece and cutting tool, and the temperature would increase as well. Therefore, the relationship between the surface roughness and processing temperature was investigated. It was found that the surface roughness was roughly positively related with temperature as was displayed in Figure 5, which indicated that temperature might have greater effects on the material removal process.

As shown in Figure 5, the blue line is the linear regression results of surface roughness and the processing temperature. In general, the surface roughness increased with the increasing of processing temperature. For example, the surface roughness is about 45 nm when the temperature is around 70 °C. When the temperature rises to about 120 °C, the surface roughness will increase to about 180 nm. The processing temperature is the comprehensive result of spindle speed, depth of cut, and feed rate. In addition, the processing temperature influences the machining quality and proper machining parameters can be chosen to achieve better surface quality. Further study such as dynamic property and cutting chip characteristics should be done to clarify how the temperature influences the material removal process. According to the results of DMA experiments, when the temperature is in the range of 60–70 °C, the PMMA is in the glass state under the circumstances and the material can be removed in the brittle removal method. In this temperature range of 70–120 °C, the PMMA is in the viscoelastic state and the material cannot be removed brittlely. The higher temperature causes the chips to burn and form amorphous chips. During the whole milling process, the obtained temperature is not higher than 120 °C, and this indicates that the material is not transferred into the rubbery state.

### 3.3. Thermal Effect on Cutting Chips and Burrs

#### 3.3.1. The Classification of Cutting Chips

The cutting chips were classified into three types according to their shapes as is shown in Figure 6: cutting chip in a roll shape, cutting chip in a sheet shape, and cutting chip in a sinter shape. The size of rolled chips could not be measured directly as displayed in Figure 6a. The length of the chip was calculated indirectly by the calculation method of the helix and the helix angle β was between 30–60 °C. According to the results, cutting chip length was in the range of 500–650 μm whose width roughly equaled the cutting depth. In Figure 6b, the length and width of cutting chips in sheet shape could be measured directly with LSCM. It was found that the chip width and cutting depth were approximately equal to each other. Sinter cutting chips were presented in Figure 6c, and they are amorphous and agglomerated into variable size.

#### 3.3.2. The Classification of Burrs

The grove edge morphology is presented in Figure 7 and classified into four types according to the edge burrs’ characteristics: Neat Edge, Small Burrs, Big Burrs, and Sintered Edge. For “Neat Edge” grooves, the edges were neat and almost no burrs could be found under the microscope. Some small burrs could be observed for the “Small Burrs” edge, and the groove bottom quality was as good as that of the “Neat Edge”. The macro burrs were visible even without a microscope for the “Big Burrs” edge, and the groove bottom surface quality decreased. For the “Sintered Edge” groove, the edges were irregular, and large numbers of burning burrs were observed.

#### 3.3.3. Influence of Temperature on Surface Quality

The burrs’ characteristics under different milling parameters were listed in Table 2. Based on this table, it can be found that the quality of groove edges depends on three factors, spindle speed, feed rate, and depth of cut. In most cases, the sintered edge and edge with small burrs appeared in the same groove with a feed rate of 5 mm/s or 9 mm/s. With the increase of feed rate, the processing temperature is increased simultaneously, which will result in the phase transition of PMMA. Under this circumstance, the samples are in the viscoelastic state, and the burrs are generated. The average processing temperature of the samples with both the sinter edge and small burrs is the highest in Table 2. The big burrs can be observed on the samples with a small feed rate and the temperature was in the range of 40–70 °C. Based on the DMA results, the material began to melt in this temperature interval. With a low feed rate, the cutting chips were very sticky and not easily removed from the sample edge. If the processing temperature kept rising, the material was burnt, and a sintered edge with small burrs was generated.

The surface roughness and temperature distribution with different groove edge morphology with various kinds of cutting chips were shown in Figure 8. As mentioned in the former part, there are four kinds of edge morphology and cutting chips in this work. In Figure 8a, the surface roughness Ra of rolled chips is between 47–173 nm, and the roughness of sheet chips is between 50–120 nm. In general, the surface roughness of the roll and sheet chips is not much different from each other. However, the bottom surface roughness Ra of sintered chips is rougher and surface roughness Ra is between 60–180 nm. Therefore, higher processing temperature results in higher surface roughness.

On the whole, with the increasing of temperature, the surface roughness kept rising as well. The average temperature of the workpiece with sintered cutting chips is about 91 °C, which is higher than the other workpieces as presented in Figure 8b. The processing temperature of the roll and sheet cutting chip is in the range of 40–80 °C. In the range of 60–80 °C, some sinter cutting chips could also be found. When the temperature was higher than 90 °C, only a sinter cutting chip can be observed. When the processing temperature was about 90 °C, the polymer was in the viscoelastic state and the material was “burned” from the sample instead of cutting from the sample.

There is a clear division between the chip morphology, the feed rate, and processing temperature as shown in Figure 9. Three regimes could be defined according to the shape of the cutting chip. Rolled chips are distributed between 40–60 °C, and sheet chips are distributed between 60–70 °C. According to the results of DMA experiments, PMMA will be in the viscoelastic state instead of the glass state if the temperature is larger than 70 °C. If the PMMA is in the glass state under the circumstances and the material can be removed in the brittle removal method, the sintered chips are distributed between 70 °C and 120 °C, and the PMMA is in the viscoelastic state under this condition. The higher temperature causes the chips to burn and form amorphous chips. During the whole milling process, the obtained temperature is not higher than 120 °C, and this indicates that the material is not transferred into the rubbery state.

## 4. Conclusions

The study analyzed the thermal effect on PMMA material removal in the micromilling process. Based on the results, the conclusions can be drawn as follows:

(1) The dynamic mechanical behavior of PMMA under different temperatures was investigated. According to the results, the PMMA glass transition temperature was 116 °C and, during the micromilling process, the sample was in the glass and viscoelastic state. In the glass state, the material is removed brittlely with better bottom surface roughness and neat edge.

(2) The morphology of groove edges is classified into four types according to the edge burrs’ characteristics: Neat Edge, Small Burrs, Big Burrs, and Sintered Edge. The sintered edge with the small burrs obtained the highest average processing temperature (115 °C). With the increasing of temperature, the material will be burned from the workpiece, which will result in big burrs or a sintered edge instead of being cut away directly. 

(3) The cutting chips are classified into three different kinds according to their shapes: roll, sheet, and sintered. The surface roughness of samples with sheet and roll cutting chips is smaller than that of sintered cutting chips. To obtain a better machining bottom surface and edge shape, a processing temperature under 70 °C is proposed according to the results.

In general, processing temperature affected the machined quality in the micromilling process. According to the results, the viscoelastic response of PMMA is sensitive to the temperature according to the DMA results. During the micromilling process of PMMA, the temperature ranged from about 40 °C to 120 °C, and the material changed from the glassy sate to the viscoelastic state, which leads to different kinds of burrs and cutting chips. Further study should be done considering the thermal effect on the constitutive property and another essential mechanical property to improve the quality of ultra-precision machining.

## Figures and Tables

**Figure 1 polymers-12-02122-f001:**
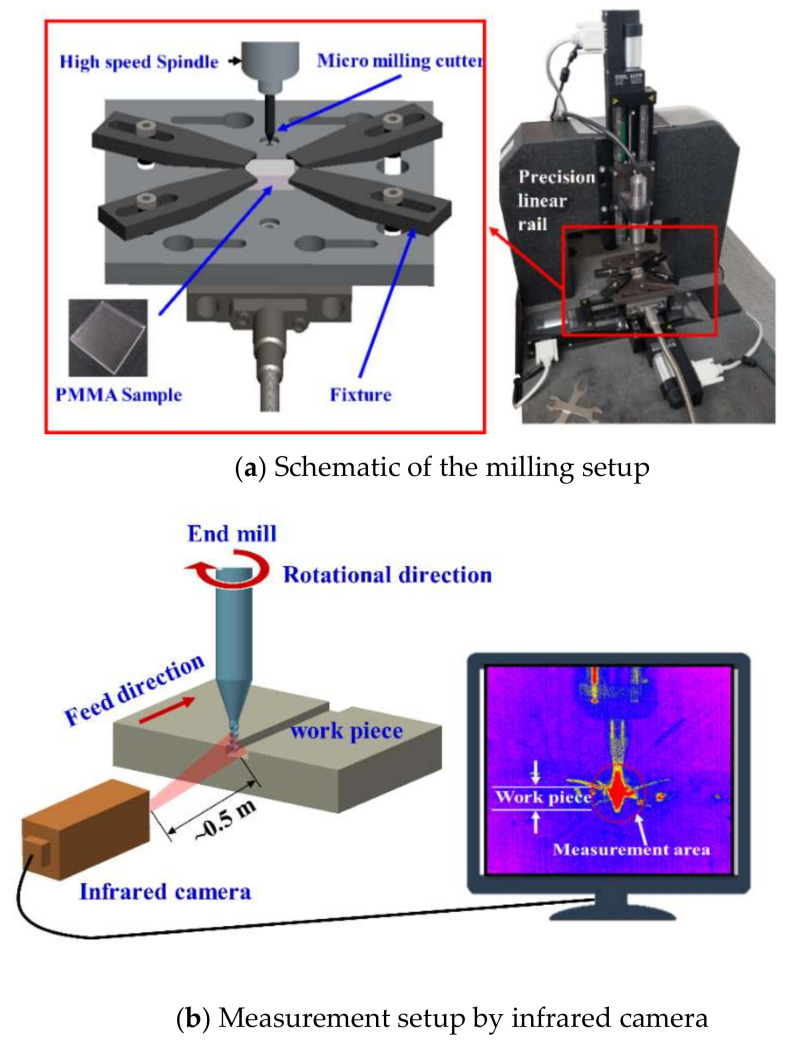
Sample and schematic of the experimental setup.

**Figure 2 polymers-12-02122-f002:**
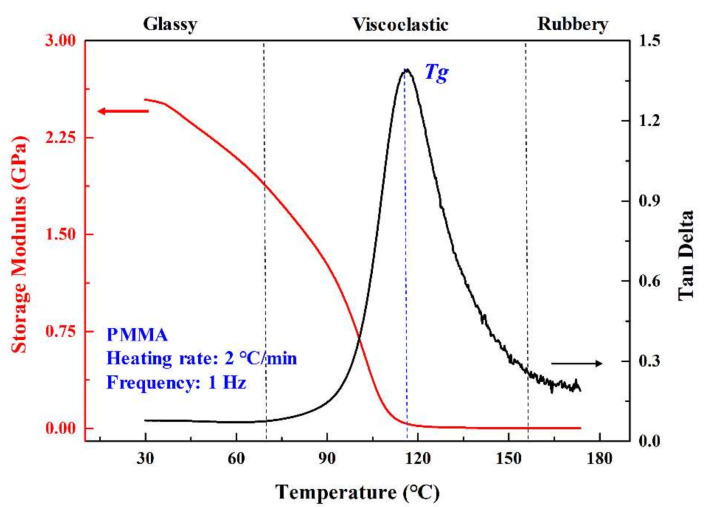
Dynamic properties (modulus and tan delta) vary with temperature (Sample: PMMA, Temperature range: 30–180 °C, Temperature step: 2 °C/min, Frequency: 1 Hz).

**Figure 3 polymers-12-02122-f003:**
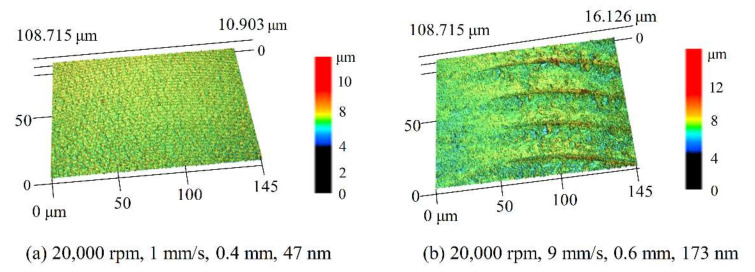
Grove bottom surface morphology with different processing parameters: spindle speed-feed rate-cutting depth-surface roughness (**a**) 20,000 rpm, 1 mm/s, 0.4 mm, 47 nm; (**b**) 20,000 rpm, 9 mm/s, 0.6 mm, 173 nm. (The results were measured by the laser scan confocal microscope (VK-X250, Keyence Co., Osaka, Japan) with a 20× microscope lens).

**Figure 4 polymers-12-02122-f004:**
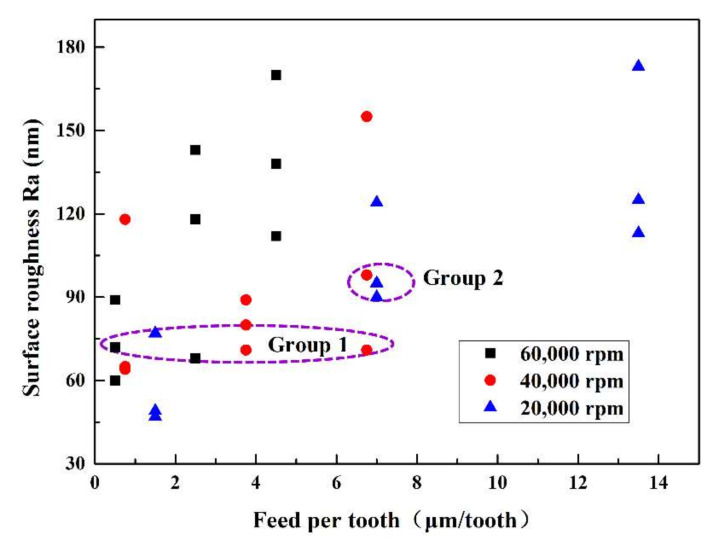
Grove bottom surface morphology with different processing parameters. (The results were measured by the laser scan confocal microscope (VK-X250, Keyence Co., Osaka, Japan) with a 20× microscope lens).

**Figure 5 polymers-12-02122-f005:**
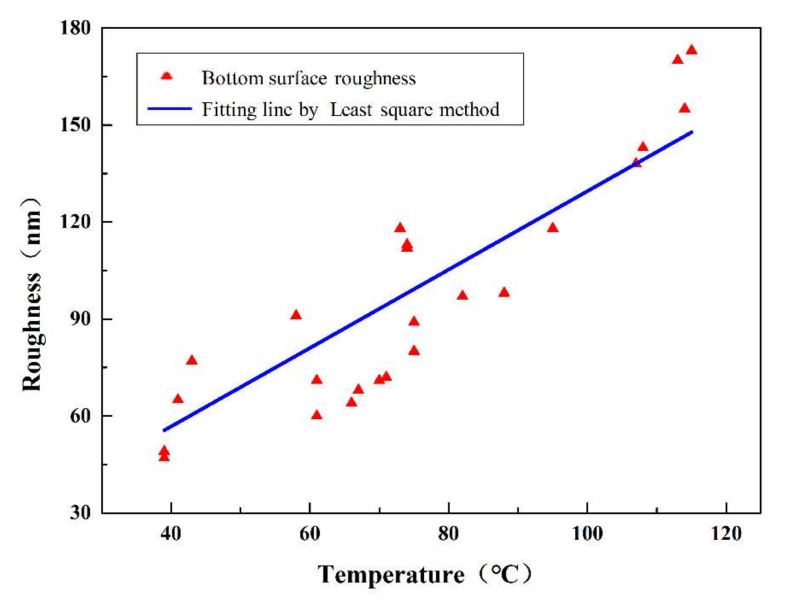
Relationship between processing temperature and groove surface roughness. (The temperature was measured during the micro milling experiments with the parameters in Table 1 by the infrared camera (IGA 6 Advanced, Lumasense Technologies Co., Stockholm, Sweden)).

**Figure 6 polymers-12-02122-f006:**
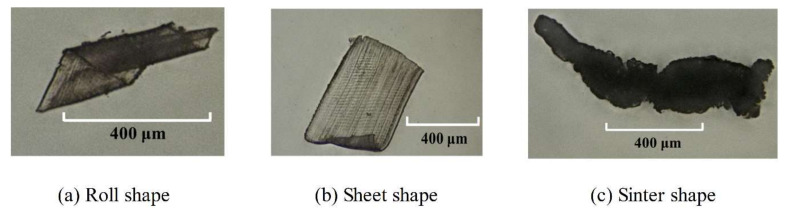
Cutting chips in different shapes: (**a**) roll shape; (**b**) sheet shape; (**c**) sinter shape. (The cutting chips were collected after the micromilling experiments with parameters in Table 1 and measured by the laser scan confocal microscope (VK-X250, Keyence Co., Osaka, Japan) with a 10× microscope lens).

**Figure 7 polymers-12-02122-f007:**
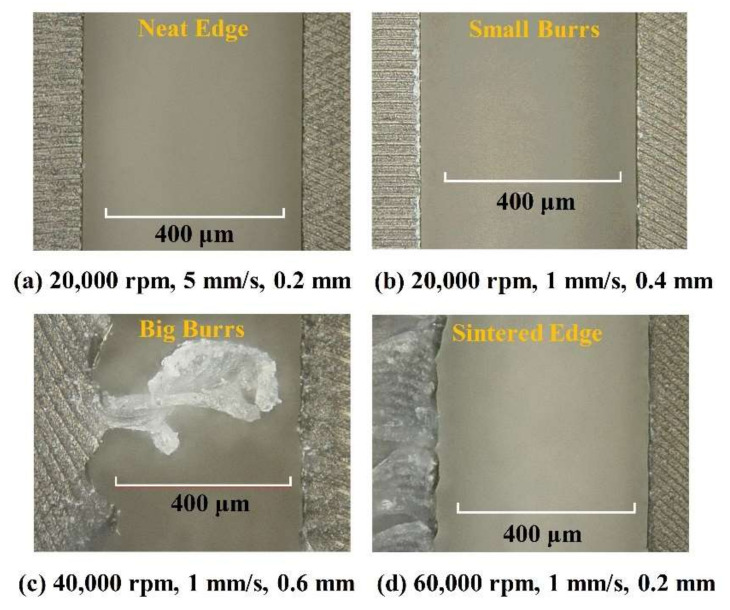
Grove edge morphology classification. The grooves were machined by different processing parameters, spindle speed-feed rate-cutting depth: (**a**) 20,000 rpm, 5 mm/s, 0.2mm; (**b**) 20,000 rpm, 1 mm/s, 0.4 mm; (**c**) 40,000 rpm, 1 mm/s, 0.6 mm; (**d**) 60,000 rpm, 1 mm/s, 0.2 mm. (The results were measured by the laser scan confocal microscope (VK-X250, Keyence Co., Osaka, Japan) with a 10× microscope lens).

**Figure 8 polymers-12-02122-f008:**
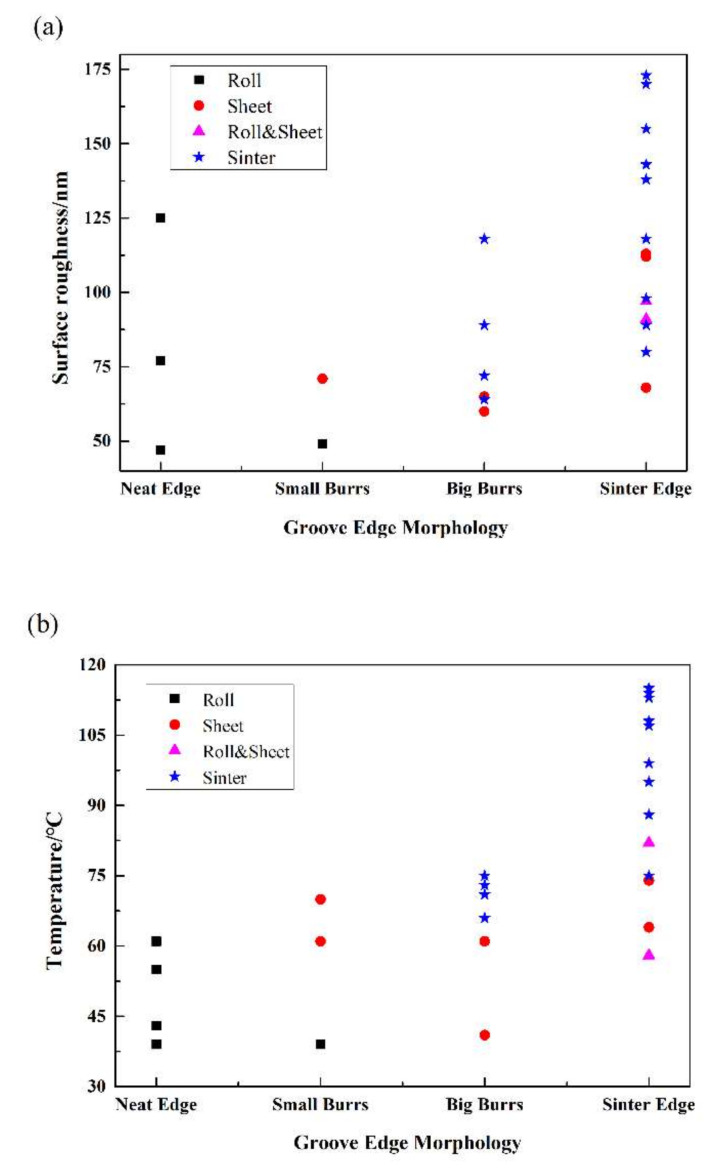
(**a**) surface roughness distribution with different groove edge morphology and cutting chips in various shapes; (**b**) temperature distribution with different groove edge morphology and cutting chips in various shapes. (The surface roughness and processing temperature were the results of micromilling experiments with parameters in Table 1 and measured by the infrared camera (IGA 6 Advanced, Lumasense Technologies Co., Stockholm, Sweden) and the laser scan confocal microscope (VK-X250, Keyence Co., Osaka, Japan) with a 10× microscope lens).

**Figure 9 polymers-12-02122-f009:**
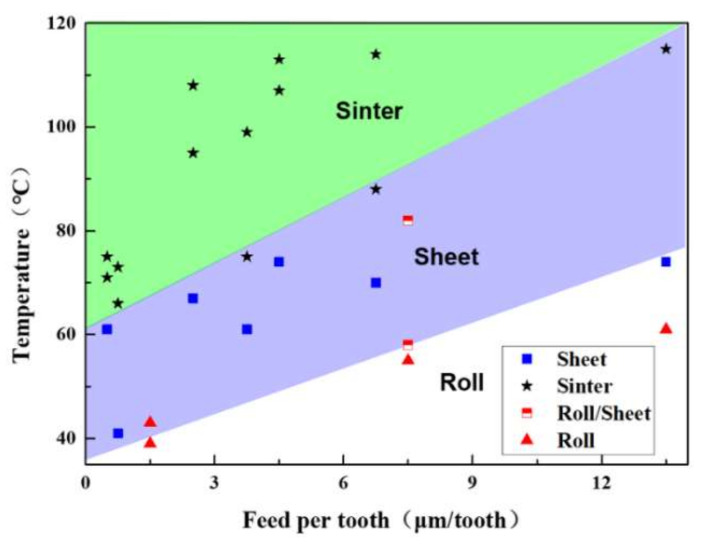
Cutting chip regimes as a function of feed per tooth and temperature. Three cutting chip regimes are identified: roll shape regime, sheet shape regime, and sinter shape regime. (The processing temperature were the results of micromilling experiments with parameters in Table 1. The cutting chip shape and temperature were measured by the laser scan confocal microscope (VK-X250, Keyence Co., Osaka, Japan) with a 10× microscope lens and the infrared camera (IGA 6 Advanced, Lumasense Technologies Co., Stockholm, Sweden), respectively).

**Table 1 polymers-12-02122-t001:** Micromilling parameters for temperature effect.

Factor	Level 1	Level 2	Level 3
Spindle speed (rpm)	20,000	40,000	60,000
Feed rate (mm/s)	1	5	9
Depth of cut (mm)	0.2	0.4	0.6

**Table 2 polymers-12-02122-t002:** Burrs’ characteristics under different micromilling parameters.

Spindle Speed (10^4^ rpm)	Feed Rate (mm/s)	Depth of Cut (mm)	Neat Edge	Small Burrs	Sinter	Big Burrs	Temperature (°C)
2	5	0.2	✓				55
2	9	0.2	✓				61
2	1	0.4	✓				39
2	1	0.6	✓				43
2	1	0.2		●			39
4	5	0.2		●			61
4	9	0.2		●			70
2	5	0.4			▲		58
2	9	0.4			▲		74
2	5	0.6		●	▲		82
2	9	0.6		●	▲		115
4	5	0.4		●	▲		75
4	9	0.4		●	▲		88
4	5	0.6		●	▲		99
4	9	0.6		●	▲		114
6	9	0.2		●	▲		74
6	5	0.5		●	▲		67
6	5	0.4		●	▲		95
6	9	0.4		●	▲		107
6	5	0.6		●	▲		108
6	9	0.6		●	▲		113
4	1	0.2				★	41
4	1	0.4				★	66
4	1	0.6				★	73
6	1	0.2				★	61
6	1	0.4				★	71
6	1	0.6				★	75

There are four kinds of burrs’ characteristics, four kinds of symbols represent the groove has the characteristics. (Neat Edge: ✓, Small Burrs: ●, Sinter: ▲, Big Burrs: ★).

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
