# Peer review of "Thermal Effect on Poly(methyl methacrylate) (PMMA) Material Removal in the Micromilling Process"

_polymers, 2020, doi:10.3390/polym12092122_

Round 1

Reviewer 1 Report

Manuscript entitled “Thermal Effect on PMMA Material Removal in 2 Micromilling Process” has been carefully read and analysed in context of publication in the journal Polymers.

In my opinion the manuscript is well written and presents decent level of expertise in the area of polymer processing, so I suggest to accept it after minor revision.

Below you can find detailed opinion about reviewed article.

Abstract of the article is reasonable and provide with necessary information. The Introduction of the manuscript consists of 5 paragraphs providing with necessary basic information. However, I have found that in line 26 chemical name of PMMA is not written according to IUPAC, it should be poly(methyl methacrylate). later in line 38 carbon dioxide formula should contain 2 in subscript.

The goal of the study provided in last paragraph of the introduction should be improved. It should contain detailed articulation of aim of the study including the explanation what authors are going to achieve and wat is the scientific value of the study. The choice of PMMA as the study object seem reasonable in the context of preceding paragraphs of introduction but it would be valuable if authors could point out some arguments that show their work in specific context of the polymer materials machinability.  

Results and discussion is well written, however it could be improved in following points:

  1. On Fig. 2. two plot are presented each of them have different OY axis, however there is no unambiguous indication to which axis the curve is related. Authors should place arrows next to each plot pointing to the right axis.
  2. The caption of Fig. 2. should include the name of investigated polymer and measurements details (conditions).
  3. Table 2 need correction with content alignment.

The Conclusion summarizes most important findings of the study. Nevertheless it does not provide the clue what is the study contribution to deeper understanding of polymer surface quality influence factor and material removal mechanism signaled in the introduction.

In my opinion improvement of the issues raised in my review will make the manuscript worth publishing in the journal Polymers.

Reviewer 2 Report

The manuscript deals with the study of physical properties of polymethyl methacrylate (PMMA) under micromilling processing. This polymer is used in several applications such as microfluidics and the performance of well controlled and shaped PMMA materials is the motivation of the present research. In this work, the authors investigate the effect of temperature during the polymer micromilling process and the dynamic mechanical analysis (DMA) approach was used to study the mechanical properties of the materials.

The state of the art is consistent and references are representative of the topic. The overall experimental design is quite clear, even thought the research is more routinary. The results are reported in a classificatory way, without an effort to explain these data in terms of some models or theoretical interpretations.

For example, about the thermal effects investigation, the data are quite confusing. In particular, what is the meaning of Figure 4? Data are very spread and it seems that no trends are present. This point should be clarify in order to justify and explain these data. Moreover, in Figure 5 a line is drawn but it is not clear if this is the results of a linear regression and what is the meaning of the involved parameter. Is this curve a line? Are the linear parameters related to some physical properties?

I suggest the authors to improove the interpretation of the data and the discussion of the results.
